# Brightness Prediction of Large Color Gamut Laser Display Devices

**DOI:** 10.3390/mi14101850

**Published:** 2023-09-27

**Authors:** Jianying Zhu, Weinan Gao, Yong Bi, Zuyan Xu, Minyuan Sun

**Affiliations:** 1Applied Laser Research Center, Technical Institute of Physics and Chemistry, Chinese Academy of Sciences, Beijing 100190, China; zhujianying18@mails.ucas.ac.cn (J.Z.); biyong@mail.ipc.ac.cn (Y.B.); zyxu@mail.ipc.ac.cn (Z.X.); sunminyuan@mail.ipc.ac.cn (M.S.); 2University of Chinese Academy of Sciences, Beijing 100049, China

**Keywords:** laser display, brightness, H-K effect, BT.2020

## Abstract

A brightness-perceived color appearance model tailored for large gamut display devices, exemplified by laser displays, was investigated. Psychophysical experiments on the brightness matching of 30 color stimuli with achromatic white light were conducted by 16 observers. The analysis compares the performance of a number of existing color appearance models and equivalent luminance models in predicting brightness. None of the models performed acceptably due to a severe underestimation of the Helmholtz–Kohlrausch (H-K) effect. A modified model of perceived brightness based on CAM16, taking into account the H-K effect, is proposed. Evaluated by psychophysical experiments, the proposed model exhibits a superior performance compared to the preceding models, especially within the extensive color gamut range stipulated by BT.2020. The results help to optimize the design of laser displays with a wide color gamut and high perceived brightness.

## 1. Introduction

Over an extended period, the development of display devices, positioned as pivotal information output interfaces within the realm of human–computer interaction, has garnered significant attention. The current trajectory of display technology is predominantly characterized by the pursuit of ultra-high resolution, expansive dimensions, and vibrant color rendition. In contemporary times, a diverse array of emerging display technologies is on the horizon, exemplified by the laser display. This technology has witnessed substantial advancements in aspects encompassing resolution, color gamut range, and image quality [1]. Laser display represents a novel technological paradigm featuring a three-color (or multi-color) ensemble of red (R), green (G), and blue (B) lasers as its light source. Through the precise modulation of the intensity ratios, total luminosity, and the spatial distribution of these lasers, the technology achieves the presentation of high-quality color images. Notably, laser display has numerous advantages, including heightened luminance, expansive color gamut, pronounced chromatic saturation, energy efficiency, environmental compatibility, and extended operational lifespan [1,2,3].

In the information age, as digital image technology and various color media devices have become widely used, the challenge of achieving color matching and reproduction across different devices and observation environments has emerged as a pressing issue. A color appearance model mathematically converts CIE 1931 XYZ tristimulus values into perceptual color attributes based on observational conditions [4,5,6]. It has the ability to theoretically guide color gamut mapping and improve cross-media image reproduction. In addition, it can also explain the complex color phenomena that cannot be explained via the traditional colorimetry, such as the Hunt effect and the Helson–Judd effect [6]. Applying the color appearance theory to laser display technology can more fully exploit the advantages of the laser display’s wide color gamut to achieve superior image reproduction.

One of the most classic color appearance models is CIECAM02, which was published by the International Commission on Illumination (CIE) Technical Committee in 2002 [5]. Owing to its robust predictive performance and simplicity of use, it is widely employed in both scientific research and industrial applications [5,6,7]. In 2017, Li et al. revised the CIECAM02 model to produce CAM16. This revision merged the chromaticity adaptation transform with the cone response transform and introduced a two-step chromaticity adaptation process [8]. At present, both CIECAM02 and CAM16 are employed to predict color appearance under various conditions. Nevertheless, both models exhibit certain limitations in practical applications. Multiple studies have indicated that these models tend to underperform in luminance prediction for large gamut displays, largely because they do not account for the H-K effect [9,10,11].

The H-K effect refers to the color appearance phenomenon where highly saturated colors appear brighter than less saturated colors, even if they possess equal luminance [12,13,14]. Early display devices typically had a limited color gamut range, often not even meeting the sRGB standard. The performance of CIECAM02 was notably commendable in this context. Nowadays, with the continuous development of new display technologies, the color gamut is becoming wider and wider. Consequently, integrating the H-K effect into color appearance systems has become indispensable. Withouck et al. revised the CAM97u model to CAM97um, enhancing the color contribution in the luminance attribute to accommodate the H-K effect [9]. Kim et al. used a psychophysical experiment using an active-matrix LCD (AMLCD) monitor panel with a 98% color gamut size of the sRGB standard and proposed a correction to CIECAM02 to compensate for the H-K effect [10]. Luke Hellwig et al. extended CIECAM02 with CAM16 by correcting the computational formulas based on the existing datasets [11]. However, the color gamut of most of the display devices nowadays has exceeded the sRGB standard, and the latest color gamut standard is BT.2020, which has almost twice the color gamut area of sRGB. Due to the good monochromaticity of the laser light source, the three primary color points of the laser display are closer to the spectral color in the color gamut map. Therefore, laser display is the display technology with the widest color gamut at present, which can fully meet the requirements of the BT.2020 standard [15,16]. Consequently, it is of great significance to take laser display as the research object and propose the brightness prediction model that is more widely applicable to the display devices with a large color gamut.

This study explores the reasons for the discrepancy between perceived brightness and luminance and develops psychophysical experiments accordingly. Sixteen observers matched the brightness of 30 self-luminous colored stimuli to achromatic white light on the same dark background. The spectral radiance of the stimuli was measured, as well as the chromatic properties. This was used as a basis for assessing the performance of previous brightness models. A modification in brightness prediction is proposed based on the CAM16 color appearance model, taking into account the H-K effect. The proposed new model will provide theoretical guidance for the development and design of new laser displays.

## 2. Analysis

With the development of color science, traditional colorimetry has gradually shown its inadequacy in the face of complex color phenomena. A quintessential example is the H-K effect, which evidently contradicts the additivity principle of luminance in traditional colorimetry. According to Grassmann’s law of additive color mixture, when two colors of light are mixed, the luminance of the combined light equals the sum of the luminances of the individual lights, implying an inevitable increase in perceived brightness by the human eye. However, the H-K effect indicates that even if the physical luminance increases, the perceived brightness by the human eye can decrease due to a reduction in color purity. According to conventional colorimetry, luminance is calculated from the luminous efficiency function, which quantifies the sensitivity of the visual system to light as a function of the wavelength. The desired luminance L can be obtained via $L=k\int_{380}^{780}I\left(\lambda \right)V\left(\lambda \right)d\lambda$, where $V\left(\lambda \right)$ is the luminous efficiency function, and $I\left(\lambda \right)$ is the spectral radiance (power).

The calculation formula for L and Grassmann’s law of additive color mixture both demonstrate the additivity of luminance. This is also the theoretical foundation that allows trichromatic display devices to present various color images without distortion. However, due to the H-K effect, this additivity conflicts with the brightness prediction. One possible reason for this conflict is that the heterochromatic flicker photometry (HFP) method was used in the data collection for the initial original $V\left(\lambda \right)$ [17]. The method required observers to match a single wavelength of light to a reference gray stimulus while having a partial overlap between the two light patterns and flashing rapidly at a frame rate of 25 Hz. The color stimuli overlapped spatially with the achromatic stimuli and flashed alternately in time, as shown in Figure 1a. By adjusting the intensities of the two light beams until no flicker-induced change in brightness is perceived, one can conclude that their luminances match. However, Figure 1b depicts the direct brightness matching method. In this method, there is no flicker and the two light sources are spatially separated. The intensity of the two light sources can be adjusted separately until they are perceived as having the same brightness.

Therefore, the direct brightness matching method shown in Figure 1b will be used for the subsequent psychophysical experiments. The aim is to obtain experimental data that includes the H-K effect factor.

## 3. Experimental Setup

In this study, we used a trichromatic laser projector developed in-house as the stimulus source. The color gamut of the laser projection was measured via a Minolta CS-2000 tele-spectroradiometer in a dark room. The comparison of the color gamut of the laser projection devices with the sRGB standard and the BT.2020 standard is shown in Figure 2. Table 1 details the trichromatic chromaticity coordinates and compares the color gamut coverage between the laser projector and the BT.2020 standard.

From the data presented in the table, it is evident that the total color gamut coverage of the laser projection (67.91%) exceeds that of the BT.2020 standard (63.72%), underscoring the expansive color gamut of the laser projection. Nevertheless, there is a discrepancy in the primary green wavelength between the laser projection and the BT.2020 standard, which prevents the laser projection from encompassing the complete color gamut chart of the BT.2020 standard. The coverage rate of the laser projection color gamut over the BT.2020 gamut is 93.3%. It is the largest color gamut in the current study for the H-K effect and brightness perception.

Sixteen naïve observers, aged between 20 and 35 years (8 females and 8 males), participated in the psychophysical experiment. All participants had normal or corrected-to-normal vision and possessed normal color perception abilities. In the psychophysical experiments, observers were asked to assess the perceived brightness of the stimuli using a perceptual brightness matching method. Using this method, the numerical and scalable results of the measured perceived brightness can be obtained directly. At the beginning of the experiment, the observer will see a circular pattern, where the right half of the circle is the color light, and the left half of the circle is the matched achromatic stimulus. In this case, the white pattern below is used as a reference white point. The observer can use the remote control to adjust the luminance of the achromatic stimulus so that the left and right look is the same brightness and record the luminance of the matched achromatic stimulus as the perceived luminance of the colored light. The observer stands 450 cm away from the pattern. The circular pattern provides a field of view (FOV) of approximately 2.5° to the observer. Figure 3 shows several of the test patterns cast by the laser projector. Figure 4 illustrates the distance of the observer in relation to the observation pattern and the surrounding observation environment. The observer needs to adapt to the dark observation conditions for at least 5 min before the experiment.

To achieve a precise evaluation of the perceived brightness across the broadest possible color gamut, we meticulously curated a collection of thirty color stimuli. These stimuli were chosen to exhibit varying levels of luminance and were strategically dispersed across the entirety of the projection device’s color gamut. The distribution of these stimuli on the color gamut map is visually represented in Figure 5. Additionally, Table 2 also provides detailed information regarding the luminance and chromaticity coordinates of the observed colors, as well as the average matched brightness values from the 16 observers.

The coefficient of variation (CV) is often used to compare the agreement between two sets of data [9,18].
(1)$$\mathrm{CV}=100\sqrt{\frac{1}{n}\overset{n}{\underset{i=1}{\sum }}\frac{{\left(Q_{geom,i}-fQ_{obs,i}\right)}^{2}}{{\bar{Q}}_{geom}^{2}}}$$
(2)$$f=\frac{\sum_{i=1}^{n}Q_{geom,i}Q_{obs,i}}{\sum_{i=1}^{n}Q_{obs,i}^{2}}$$
where $Q_{obs,i}$ represents the individual observer brightness of stimulus i, $Q_{geom,i}$ is the geometric mean of $Q_{obs,i}$ for all the observers, ${\bar{Q}}_{geom}$ is the arithmetic mean of $Q_{geom,i}$ for all stimuli, n is the number of evaluated stimuli, and f is the factor adjusting the $Q_{geom,i}$ and $Q_{obs,i}$ values to the same scale.

Table 3 summarizes the results of observer consistency in the form of CV values. The results show that the interobserver mean of 15.9% is similar to the previous experiments [9,18], indicating good agreement.

## 4. Model Performances

In recent studies concerning the H-K effect and perceptual brightness, numerous models have been developed. Two commonly used classes of models were selected to evaluate their brightness prediction performance on the wide color gamut laser projection device.

### 4.1. Equivalent Luminance According to Nayatani

Nayatani proposed two methods that take into account the H-K effect for calculating the equivalent luminance [12]: the variable achromatic color (VAC) and the variable chromatic color (VCC) method, given by Equations (2) and (3), respectively:(3)$$\gamma_{VCC}=\frac{L_{eq}}{L}=0.4462{\left\{1+\left[-0.8660q\left(\theta \right)+0.0872K_{Br}\right]\times s_{uv}\left(x,y\right)+0.3086\right\}}^{3}$$
(4)$$\gamma_{VAC}=\frac{L_{eq}}{L}=0.4462{\left\{1+\left[-0.1340q\left(\theta \right)+0.0872K_{Br}\right]\times s_{uv}\left(x,y\right)+0.3086\right\}}^{3}$$
(5)$$q\left(\theta \right)=-0.01585-0.03017cos\theta -0.04556cos2\theta -0.02667cos3\theta -0.00295cos4\theta +0.14592\;sin\theta +\;0.05084sin2\theta -0.01900sin3\theta -0.00764\;sin4\theta$$
(6)$$K_{Br}=0.2717\times \frac{6.469+6.362L_{a}^{0.4495}}{6.469+L_{a}^{0.4495}}$$
(7)$$s_{uv}\left(x,y\right)=13{\left[{\left(u^{\prime }-u_{\prime }^{w}\right)}^{2}+v^{\prime }-v\prime_{w}{}^{2}\right]}^{1/2}$$

The function $\;q\left(\theta \right)$ describes the effects of the hue angle *θ* on the H-K effect. $K_{Br}$ illustrates the increase in the H-K effect when the adapted luminance of the chromatic object color is increased. The computational model of VAC was chosen here because the experiment took the approach of adjusting the luminance of the achromatic stimuli for matching.

### 4.2. CIECAM02/CAM16

CAM02 and CAM16 are the commonly used color appearance models. There is no difference between the two after the achromatic stage. Therefore, the two are treated here as a unified model. The calculation process of CAM02 is relatively complex; for the detailed calculation process of CAM02 and CAM16, please refer to the reference [5,8]. In addition, some models consider incorporating the factors of the H-K effect into the existing color appearance systems [10,11]. Among them, the Hellwig model is an extended model based on CAM16 considering the H-K effect. It has demonstrated an excellent performance in certain datasets [11]. The Hellwig model uses the form of Equation (8) to add the contribution with the H-K effect to the new $J_{HK}$, subsequently obtaining the brightness Q.
(8)$$J_{HK}=J+f\left(h\right)\cdot C^{\gamma }$$

Here, we select both CAM16 and the Hellwig model for the brightness prediction performance evaluation. Figure 6 shows the comparison of the perceived brightness matching values of the above three models in the psychophysical experiments. An ideal model predicts equal lightness (black line) for the chromatic stimuli and the experimentally-matched achromatic stimuli. If a point falls below the line, this indicates that the model underpredicts the H-K effect for that chromatic stimulus—its predicted brightness is less than the predicted brightness of the experimentally matched achromatic stimulus.

The definition of CV is consistent with Equations (1) and (2). The model calculation results are taken as $Q_{obs,i\;}$, and the average perceived brightness from the psychophysical experiments is taken as $Q_{geom,i}$. As can be seen in Figure 6, the performance of the three models is rather unsatisfactory with large CV values. The VAC model exhibits an inferior0020performance. As observed in Figure 6b,c, for the high luminance stimuli, CAM16 and the Hellwig model tend to systematically underestimate the brightness of the colored stimuli. This suggests that CAM16 does not account for the H-K effect. The Hellwig model incorporates the H-K effect into CAM16 to achieve a better brightness prediction performance for larger color gamuts. However, the color gamut range of his experimental data is only similar to sRGB, resulting in limited corrective effects when applied to wide color gamut laser display devices.

### 4.3. Proposed Model

Assuming an additive contribution of chromaticity or chroma to the lightness or brightness is a better way to explain the H-K effect. Therefore, the common correction method is to consider the H-K effect in the lightness equation and then derive the H-K compensated brightness from the lightness. One of the widely referenced forms is Fairchild and Pirrotta’s model, although it was used in CIELAB [19]. Kim introduced it to CIECAM02 and achieved good results in the sRGB color gamut [10]. Based on past research [10,11,19], it can be assumed that the new model has the following form:(9)$$J_{HK}=J+f_{1}\left(J\right)f_{2}\left(h\right)\cdot C^{\gamma }$$
where *J* is lightness, $f_{1}\left(J\right)$ represents the effect of lightness on the H-K effect, and $\;f_{2}\left(h\right)$ is the hue angle dependency. It is treated as a factor related to the hue angle, although CAM16 has claimed to have eliminated the effect of the hue angle in chromaticity. And, *C* is the chroma. The H-K-effect-compensated brightness $Q_{HK}$ is calculated from $\;J_{HK}$ by:(10)$$Q_{HK}=\left(\frac{4}{c}\right){\left(\frac{J_{HK}}{100}\right)}^{0.5}\left(A_{w}+4\right)*F_{L}^{0.25}$$

The form of $f_{1}\left(J\right)$ is assumed to have the following form:(11)$$f_{1}\left(J\right)=\alpha_{1}+\alpha_{2}J$$

$f_{2}\left(h\right)$ was hypothesized to have the following form:(12)$$f_{2}\left(h\right)=\alpha_{3}+\alpha_{4}\sqrt{\frac{1-\sin h}{2}}$$

The parameters γ and $\alpha_{1}$ to $\alpha_{4}$ are obtained after conducting a multivariate nonlinear regression on the experimental data using MATLAB. The aim is to minimize the difference between the model output $Q_{HK}$ and the average perceived brightness observed by the participants. The fitting results yielded the following values:(13)$$J_{HK}=J+f_{1}\left(J\right)f_{2}\left(h\right)\cdot C^{0.867}$$
(14)$$f_{1}\left(J\right)=1.52-0.013J$$
(15)$$f_{2}\left(h\right)=0.65+0.11\sqrt{\frac{1-\sin h}{2}}$$

Figure 7 shows the fit of our proposed model to the data from the psychophysical experiment. In this case, the CV is 15.5%, which is close to the mean value of 15.9% for the observers within the psychophysical experiment. This demonstrates its ability to predict the brightness well over a large color gamut.

## 5. Discussion

Generally, color models require the experiments to be conducted separately in dark, dim, and average environments to be applicable across most scenarios. Since projection devices use reflective imaging, excessive ambient light reflecting off the screen and into the eyes can impair the viewing experience and also affect the reliability of the experimental data. Hence, we conducted color matching experiments solely in a dark environment. Moreover, the differences among the three environments are reflected in the three environmental parameters in the model. Using the parameters from CAM16 directly, the resulting errors are not significant.

It should be noted that the H-K effect is not only affected by color saturation, but also changes with brightness. That is, the lower the luminance, the more pronounced the H-K effect is, hence the reason why several sets of color stimuli with different color saturation and luminance were selected for the experiment. Due to the visual characteristics of the human eye, in the general three-color display equipment, the peak luminance of blue is generally much lower than the remaining two colors. The laser projector used in the experiment, for example, the peak luminance of its blue light, is only about 30 nit, which is much lower than the green at 220 nit. This is also the reason why the color spikes could not be selected for the experiment with the same luminance. Evaluating the luminance perception of complex images and extending the luminance range to meet the HDR conditions are also issues to be addressed in the future.

On the other hand, the good coherence of the laser brings some scattering noise that affects the quality of the displayed picture. To overcome this challenge, we adopted a multi-wavelength design, sacrificing a bit of the color gamut, and developed an 8K laser display prototype. Compared to the laser projector used in the previous experiments, it has a slightly smaller color gamut range, higher resolution, and reduced speckle noise. However, due to some other issues, its maximum luminance is only 175 nit. Nonetheless, compared to traditional display devices, it utilizes a lower physical luminance to achieve an impressive perceived brightness effect. Table 4 shows the three primary chromaticity coordinates of the new laser display as well as a percentage of the full color gamut. Figure 8 shows the coverage on the color gamut chart. Figure 9 shows a physical drawing of the prototype.

At the same time, several observers were invited to view the same images using the new 8K laser TV and a conventional LCD display. The observers could adjust the screen brightness to a range they felt comfortable with on their own. The actual physical luminance of the two displays was then compared using the cs-2000. Statistical results indicate that, when the observers perceive the luminance as comfortable, the actual physical brightness of laser display devices is 25% to 40% lower than that of LCD devices. This suggests that the H-K effect, induced by the wide color gamut display devices, is notably prominent in complex visual stimuli. The quantification of the H-K effect in complex images is the subsequent step that needs to be undertaken.

## 6. Conclusions

This study invited 16 observers to conduct psychophysical experiments on brightness matching using laser projection over a large color gamut of bt2020. The existing model performs poorly due to the prominent H-K effect in large color gamut display devices. Based on the CAM16 model, an improved model is obtained by taking the chromaticity contribution to the brightness into account. The performance of the model is more consistent with the psychophysical experimental results and proves to be more applicable to the large color gamut laser display devices. The model provides a theoretical basis for the luminance design of laser display devices. However, subsequent analyses indicate that the model’s applicability is constrained by the color gamut and the dynamic range of present-day display devices. Future implementations on devices boasting a broader color gamut and heightened dynamic range, as well as the evaluations involving intricate image models, necessitate further refinement.

## Figures and Tables

**Figure 1 micromachines-14-01850-f001:**
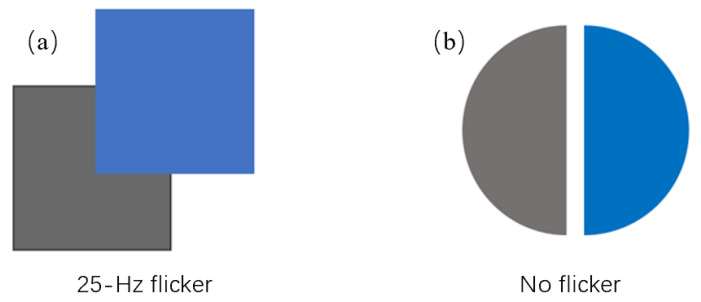
Schematic diagram of the two brightness (luminance) matching methods.

**Figure 2 micromachines-14-01850-f002:**
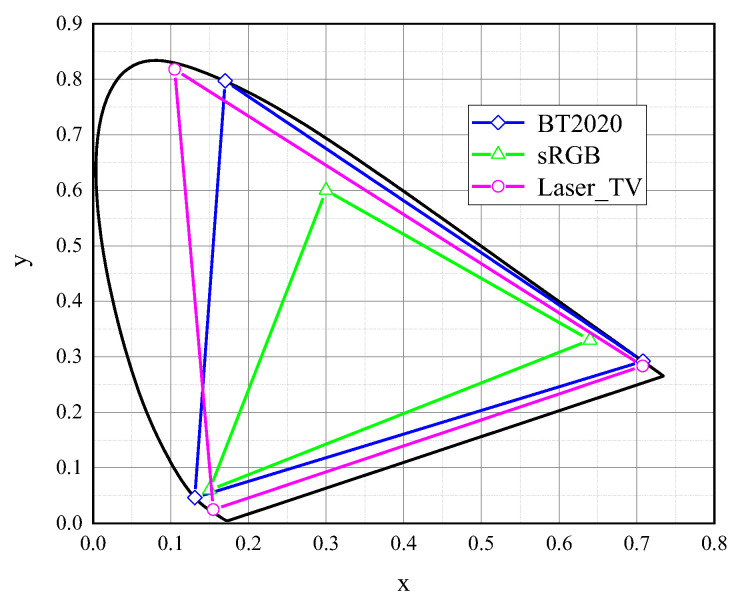
Color gamut of the laser projection plotted in x–y chromaticity diagram (CIE 1931).

**Figure 3 micromachines-14-01850-f003:**
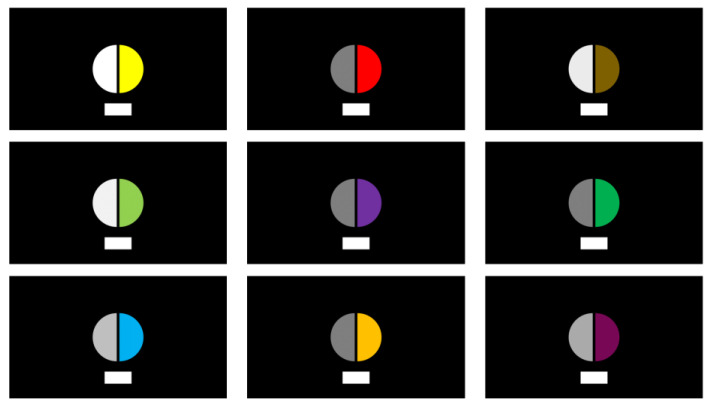
Color stimulus patterns used in psychophysical experiments.

**Figure 4 micromachines-14-01850-f004:**
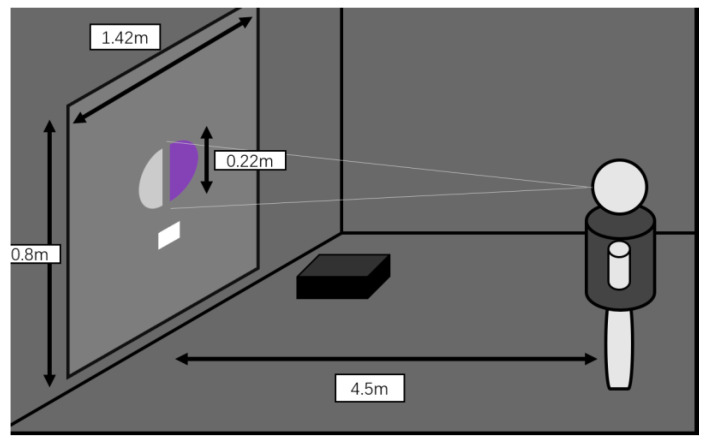
Schematic diagram of the experimental station.

**Figure 5 micromachines-14-01850-f005:**
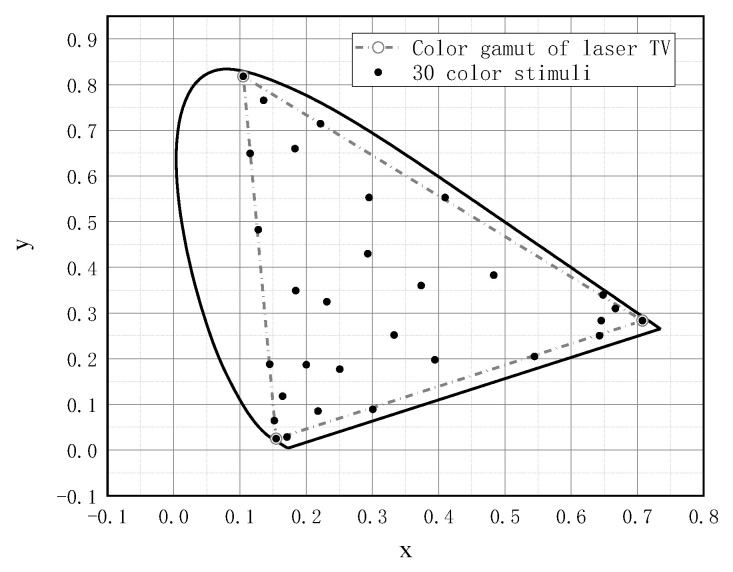
Distribution of 30 color stimuli on the color gamut map.

**Figure 6 micromachines-14-01850-f006:**
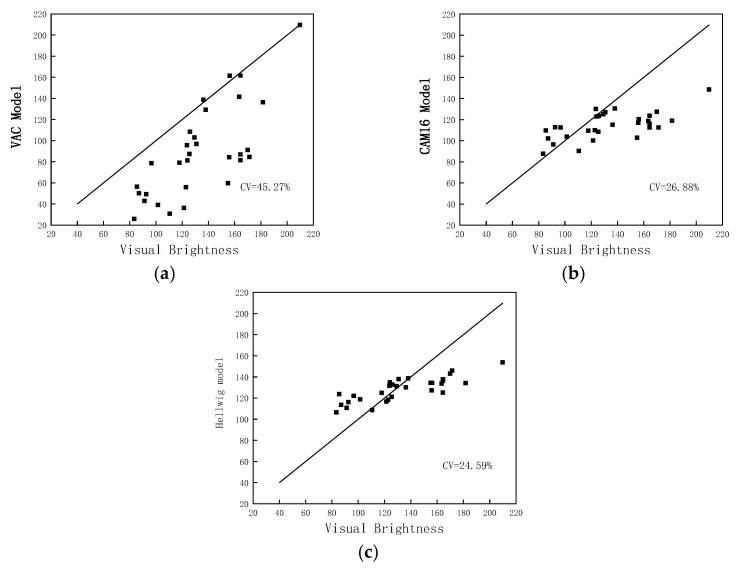
Comparison of the predictions of models on psychophysical experiment brightness matching data. (**a**) VAC model; (**b**) CAM16; (**c**) Hellwig model.

**Figure 7 micromachines-14-01850-f007:**
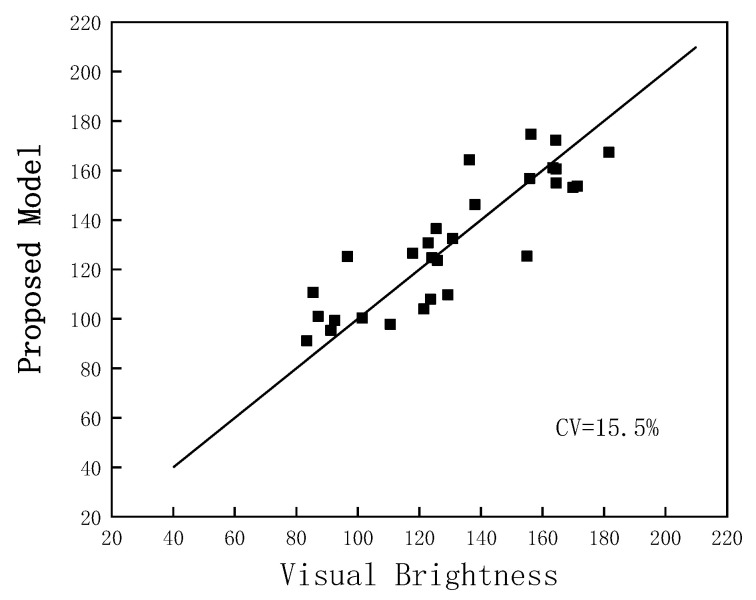
Comparison of Perceived Brightness Matching Values between proposed model and Psychophysical Experiment.

**Figure 8 micromachines-14-01850-f008:**
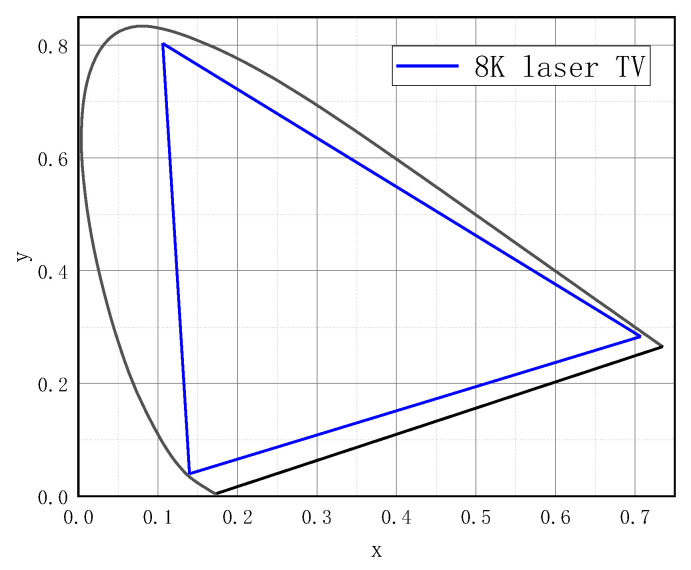
Color gamut of 8K laser TV.

**Figure 9 micromachines-14-01850-f009:**
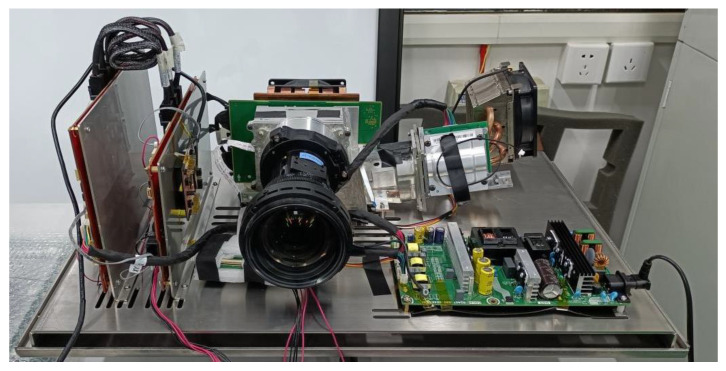
Physical drawing of the 8K laser TV prototype.

**Table 1 micromachines-14-01850-t001:** Color gamut of the laser projection to the BT.2020 standard.

	Red		Green		Blue		Gamut Coverage
	x	y	x	y	x	y	
Laser projection	0.7077	0.2834	0.105	0.8181	0.1547	0.0247	67.91%
BT.2020	0.708	0.292	0.170	0.797	0.131	0.046	63.72%

**Table 2 micromachines-14-01850-t002:** The 30 color sets used in the psychophysical experiment.

Test Colors	L	x	y	Average Perceived Brightness
1	40.49	0.7077	0.2834	164.42
2	64.6	0.105	0.818	136.27
3	9.803	0.1547	0.0247	83.42
4	47.28	0.3005	0.0891	181.60
5	189.2	0.4098	0.5527	209.76
6	93.27	0.1451	0.1876	169.95
7	32.11	0.1646	0.1179	101.45
8	19.12	0.2178	0.0852	91.22
9	26.22	0.3944	0.1975	87.07
10	46.22	0.4831	0.3827	92.50
11	52.34	0.295	0.5529	96.60
12	46.75	0.1843	0.3489	85.45
13	37.97	0.6481	0.3394	122.83
14	25.35	0.5448	0.2048	154.99
15	48.93	0.1278	0.4824	117.86
16	46.6	0.2216	0.7145	125.47
17	11.17	0.171	0.0285	110.54
18	25.09	0.152	0.0644	121.41
19	94.87	0.3736	0.3599	123.59
20	73.3	0.2002	0.1866	124.05
21	70.26	0.2507	0.1768	125.90
22	90.06	0.2312	0.3247	130.80
23	103.2	0.2929	0.4299	138.02
24	75.37	0.3328	0.2518	129.24
25	73.46	0.1155	0.6494	163.40
26	39.43	0.6429	0.2504	171.36
27	47.28	0.6454	0.2833	164.46
28	87.01	0.1831	0.6596	164.37
29	49.8	0.6668	0.3097	155.87
30	78.57	0.1357	0.7655	156.26

**Table 3 micromachines-14-01850-t003:** Evaluation of Interobserver Agreement via Calculation of the CV.

Observer	1	2	3	4	5	6
CV (%)	16.9	13.4	16.3	16.0	18.1	13.4
Observer	7	8	9	10	11	12
CV (%)	15.8	15.5	18.3	12.3	16.0	17.6
Observer	13	14	15	16	Mean	
CV (%)	20.3	13.7	14.8	16.2	15.9	

**Table 4 micromachines-14-01850-t004:** Color gamut of the 8K laser TV to the BT.2020 standard.

	Red		Green		Blue		Gamut Coverage
	x	y	x	y	x	y	
8K Laser TV	0.707	0.2834	0.106	0.8034	0.1397	0.0399	66.36%
BT.2020	0.708	0.292	0.170	0.797	0.131	0.046	63.72%

## Data Availability

The data presented in this study are available on request from the corresponding author.

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
