# Peer review of "Brightness Prediction of Large Color Gamut Laser Display Devices"

_micromachines, 2023, doi:10.3390/mi14101850_

Round 1

Reviewer 1 Report

In this manuscript, the authors report a modified CAM16-based model of perceived brightness that takes into account the H-K effect. Psychophysical experiments demonstrate that the model outperforms previous models and is more applicable to the large color gamut range of BT.2020. The theoretical background and characterizations are comprehensive. This manuscript can be accepted for publication after the following questions and comments have been addressed satisfactorily:

1.     References should be checked and unified in format, e.g., Ref. 6, 11, 17, etc.

2.     Please revised the caption of Figure 2 and Figure 3.

3.     In Figure 5, the position corresponding to test color 2 (0.105,0.818) seemed to be incorrect. Please check carefully.

4.     The X-axis coordinate in Figure 6 should be uniformed.

Language needs to be carefully improved.

Author Response

Please see the attachment response1.

Reviewer 2 Report

This manuscript investigates a brightness-aware color appearance model for large gamut display devices, with a particular focus on laser displays. Existing models have proven inadequate in accurately predicting brightness due to their underestimation of the Helmholtz-Kohlrausch effect. To address this issue, a modified model based on CAM16 was proposed, taking into account the H-K effect, and it outperformed previous models, making it more suitable for the BT.2020 color gamut.

Moreover, this research has the potential to enhance the design of laser displays, providing them with a wider color gamut and higher perceived brightness. However, there are some aspects that need improvement. The captions for Figure 2 and Figure 3 require completion, and there is a discrepancy between the intended figures; specifically, Figure 5 in section 4.2 on page seven should be Figure 6. Additionally, it is recommended that the captions for the axes of Figure 6 be standardized to improve clarity. Furthermore, chromaticity coordinates should be the accurate description, not color coordinates.

Overall, the English proficiency in the entire manuscript is quite good. However, there are a few instances where specific terminologies were misused, and some descriptions could be clearer. These areas could be improved to enhance the overall quality of the document.

Author Response

Please see the attachment response2

Reviewer 3 Report

The paper describes an experiment revealing the Helmholtz-Kohlrausch effect using an experimental laser display system. The results are compared with two existing models for the effect and an improved "model" is presented. The paper ends with the description of an 8k laser system and a comparison with a LCD-display, claiming a reduced luminance setting for the laser display.

The experiment is certainly interesting but the presentation of the results needs to be adapted. A logical question which is not handled: is the reduced luminance of the 8k system due to the H-K effect? In addition the paper contains a number of questionable statements which must be corrected or need more explanation.

A comparison is made with Nayatani's VAC model for the H-K effect and with CAM16. However, considering the references [10-11] comparing with CAM16 does not look very useful. Instead the improved model of [11] should have been considered. It would then be clear whether the laser-based system differs from systems considered previously. The authors should focus on the laser-system specifics. The fact that CAM16 is not adequate and the way to improve it is well-known by now and cannot be claimed anymore.
The authors propose a modification of CAM16 very much like the one in [11], but with a different hue function. This is however not mentioned explicitly. The hue function proposed in eq.(14) has a minimum for h=180*, whereas usually a minimum is expected for h=90° (yellow). This is difficult to understand and should be explained more in detail.

The assessment of the HFP method is confusing if not wrong: the main issue is the flickering, not the overlap, although this may also have an effect. Since the color-processing steps in the human visual system are low-pass in character these steps can be silenced by applying flicker with a sufficiently high frequency. In particular this silences the H-K effect.

In lines 265-266 it is suggested that lowering the maximum luminance of a display has a beneficial health effect, which means that watching TV is bad for the eyes. Although substantial luminance levels will undoubtedly have negative effects, a reference should be given to substantiate this claim for regular luminance levels.

Some essential details are missing:

- minimal data should be given on the test persons (age, experience, sight ... ).

- The measurement results (average perceived brightness, variation) should be added to table 2.

- Add clearly which quantities are shown in the graphs of Fig.6 (a) and (b)

- It would be informative to add the sample labels to the data points in Fig 6. This could confirm the statement in lines 205-206, which cannot be judged now.
It looks like some samples are missing in Fig.6 ?
With some imagination one can discern some subgroups in Fig.6 (a).

- What is the meaning of the CV mentioned in Fig.6; The rather small difference between (a) and (b) looks odd to me or this is not the right quantity to assess the variability.

The language is rather poor and in some instances the authors use sentences saying what is probably just the opposite of what they were trying to say (see lines 36-41).
The sentence (lines 149-152) is difficult to understand and should be reformulated.
line 154: "Thirty" should not be capitalized.
The use of "arithmetic" and "geometric" average in lines 165-166 is confusing and misleading. They are not the same. Probably the authors mean the "arithmetic" average.
It should be Grassman instead of Glassman (line 86 and line 94).

Author Response

Please see the attachment response1

Round 2

Reviewer 3 Report

Reviewer report of "Brightness prediction of large color gamut laser display devices" (micromachines-2516366 revised version).

The paper describes an experiment revealing the Helmholtz-Kohlrausch effect using an experimental laser display system. The results are compared with existing models for the effect and an "improved model" is presented. The paper ends with the description of an 8k laser system and a comparison with a LCD-display, claiming a reduced luminance setting for the laser display is overcome by the H-K effect, but this is merely a suggestion.

A comparison is made with Nayatani's VAC model for the H-K effect and with CAM16 as modified by Hellwig [11].
However I do not understand the "modification to Hellwig's model" mentioned in lines 200-201.

The model proposed by the authors is not really new. Essentially it's Fairchild's and Pirrotta's model (based on CIELAB). [Note that |sin((h-pi/2)/2)| = sqrt((1-sin(h))/2).]
M. D. Fairchild and E. Pirrotta, “Predicting the lightness of chromatic object colors using CIELAB,” Color Research & Application, vol. 16, pp. 385–393, Dec 1991.
This model was translated to CIECAM02 by Kim [10] replacing the CIELAB C* by the CIECAM02 colorfulness M. This was later questioned by Hellwig & Fairchild [11] which also dropped the J-dependent factor and used Nayatani's hue-factor instead of their original h-factor.
I think it would be fair to include the reference to the Fairchild and Pirrotta paper.

Minor changes to be made

"Luke's" should be "Hellwig's" in line 200, but maybe this should be omitted (see higher).

Reference [9] should be added in line 169.

Author Response

Please see annex  response1
